# Multi-Variant Modal Analysis Approach for Large Industrial Machine

Kajetan Dziedziech *, Krzysztof Mendrok 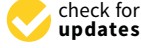, Piotr Kurowski and Tomasz Barszcz

Department of Robotics and Mechatronics, AGH University of Science and Technology, al. Mickiewicza 30, 30-059 Krakow, Poland; mendrok@agh.edu.pl (K.M.); kurowski@agh.edu.pl (P.K.); tbarszcz@agh.edu.pl (T.B.)
* Correspondence: dziedzie@agh.edu.pl

**Abstract:** Power generation technologies are essential for modern economies. Modal Analysis (MA) is advanced but well-established method for monitoring of structural integrity of critical assets, including power ones. Apart from classical MA, the Operational Modal Analysis approach is widely used in the study of dynamic properties of technical objects. The principal reasons are its advantages over the classical approach, such as the lack of necessity to apply the excitation force to the object and isolate it from other excitation sources. However, for industrial facilities, the operational excitation rarely takes the form of white noise. Especially in the case of rotating machines, the presence of rotational speed harmonics in the response signals causes problems with the correct identification of the modal model. The article presents a hybrid approach where combination of results of two Operational Modal Analyses and Experimental Modal Analysis is performed to improve the models' quality. The proposed approach was tested on data obtained from a 215 MW turbogenerator operating in one of Polish power plants. With the proposed approach it was possible to diagnose the machine's excessive vibration level correctly.

**Keywords:** Operational Modal Analysis; large industrial machine; Experimental Modal Analysis; combination of approaches



## 1. Introduction

Power generation systems are fundamental for modern societies in all countries across the world. A growing trend towards renewable energies is a global phenomenon which impacts all the other power technologies [1]. Paradoxically, the more participation of renewable energy sources there is in the system, the more reliable sources are needed to stabilize the system in low-insolation or low-wind periods [2,3]. Currently, and for the foreseeable future, such sources are thermal plants. These plants differ in fuels, resulting emissions, and the most popular are: natural gas, coal (hard and lignite), and nuclear. The reliability of thermal plants is thus a crucial factor for efficient, safe, and reliable power systems.

The core element of each thermal plant is a turbine—generator set. Such sets differ in size and power, ranging from 50 MW (except small industrial units) up to 1000 MW [4,5]. Manufacturers perform constant efforts to increase efficiency and decrease fuel consumption and emissions. Apart from many ways to achieve this goal, these mechanical systems of rotating machines are continuously improved and modernized. On the other hand, it leads to decreased tolerances, smaller gaps (e.g., between shaft and casing) and requires increased precision in operation [6]. First of all, for trouble-free operation, each unit must undergo proper alignment and balancing [7], but often these methods are not sufficient, especially if the root cause is of a different kind.

The dynamic state is of paramount importance for turbo-sets. The vibration-based analysis has become a standard procedure, and Turbine Supervisory Equipment (TSE) systems with relative shaft vibration and absolute bearing vibration are de facto standard

in assessing vibration severity [8]. Recent developments have led to introduction of a new set of standards in this field [9,10].

A much more challenging part of dynamic state assessment is the analysis of structures. Each turbine-set is a very complex mechanism which consists of numerous structures, e.g., casings, valve chambers, bearing pedestals. The overall dynamic state results from excitation from rotating shafts and the transfer path determined by the structural components. Structural analysis can help solve encountered problems which often prevent a unit from starting or achieving the full load. The most efficient and robust method for such an analysis is the Modal Analysis. In a number of publications one can find examples of the use of modal analysis to identify and monitor the operation of rotating machines in the energy industry [11–15].

The Modal Analysis can be applied in the Experimental Modal Analysis (EMA) [16–19] or Operational Modal Analysis (OMA) [20,21] form. Comparison of the two modal analysis techniques was presented in the paper by Orlowitz and Brand [22]. The former is based on the active vibration tests. That is, it requires controlled and measured excitation. The interested reader can find detailed descriptions together with mathematical formulae in given references. In this approach, it is also essential to isolate a tested system from other sources of vibration. That is why, in industrial conditions dynamic analysis is performed in most cases using OMA [23,24]. This is because shutting down the machine in plants of continuous production leads to significant financial losses. The second reason for using an OMA is the problem with proper excitation of large and stiff structures. Comprehensive review of the different OMA algorithms can be found in the article by Zahid et al. [25]. The attempt to create an automated OMA together with thorough analysis of the influence of different selection of an estimation parameters can be found in paper by Rainieri and Fabbrocino [26]. OMA has many advantages such as the use of actual excitation levels and thus the study of the machine's dynamic properties at the operating point has been conducted. Apart from this, boundary conditions are real as well. However, the analysis of results obtained through operational measurements carried out on rotating machines in particular is baffling. There is a problem with separating rotational speed harmonics from structural vibration components. The correct identification of those structural components allows the modal parameters of the tested object to be estimated. It is also important to mention that Modal Analysis can be applied in a variety of ways for reconstruction of excitation force, as it can lead to the root cause identification of a fault [27].

In some cases, the ratio of structural components to harmonic ones may reach 60 dB. The problem is significant when the natural frequencies are in proximity to the excitation frequency and its harmonics. To separate easier random and deterministic components, methods that allow the latter to be filtered out from the signal could be used.

In the literature, there are several methods for this separation. They have different properties and, thus, are used for various purposes [28]. The first method for separating deterministic and random components is Time Synchronous Averaging (TSA) [29]. This allows a periodic signal to be extracted from the signal. A fundamental frequency of this component must be known. The method guarantees good separation quality. However, it is time-consuming. A similar separation method in which the knowledge of the frequency of the extracted periodic signal is required is the Dislocation Superimposed Method (DSM) [30]. This is a modification of the Random Decrement Technique (RDT) algorithm. The next method, which may be used to separate deterministic and random components, is the Linear Prediction (LP) method [31]. In this approach, it is assumed that the signal has the character of white noise. Based on a certain number of signal samples, its deterministic part is predicted using linear Auto-Regressive (AR) models.

In 1975, Widrow et al. [32] presented the Adaptive Noise Cancelation (ANC) method which was able to separate a signal into deterministic and random components using an adaptive filter. Initially, the method was applied for telecommunications signals. Its introduction to vibrational signals was presented by Chaturvedi and Thomas in 1981 [33]. The method was later modified by Randall and Li [34] who proposed using a delayed

original signal in place of the reference signal. The method is called Self-Adaptive Noise Cancellation (SANC).

Another method for removing harmonic components from the signal is the Cepstral method [35]. The signal is presented as a cepstrum, and in this domain, it is edited to remove deterministic components. The method of removing harmonic components, used to extract structural ones, is the double resampling algorithm [36]. The signal is transformed into the order domain where the harmonic components are removed. Then it is re-transformed into the time domain with a constant sampling period. The Discrete Random Separation (DRS) algorithm presented by Antoni and Randall [37] was developed for discrete and random components separation. It has also been used for pre-processing OMA data to improve visibility of structural components [38]. All the described methods are based on the information contained in the measurement data. If a particular type of motion revealing the defect is not excited, it will not be visible in the processed results.

Authors would like to present a novel approach which combines several Modal Analysis methods. The results of two independent OMAs and EMA are combined to identify the root cause of the defect. It results in a much broader and thorough analysis than any single approach and can reduce the influence of operational excitations. It is the most comprehensive approach which the Modal Analysis can give. The presented case study was taken on a 215 MW steam turbine in one of Polish power plants. The plant reported a resonance which degraded the dynamic state of the unit. No specific solution (e.g., balancing or alignment) was successful, and the proposed extended version of Modal Analysis was chosen to identify the problem.

The paper is organized as follows: in Section 2, a description of OMA and EMA methodology are described, Section 3 describes experimental conditions and parameters, Section 4 presents the obtained results, and finally, Section 5 summarizes the paper with conclusions concerning the combination of approaches.

## 2. Description of Approach and Methods

The main idea of the proposed approach is to combine the results from two independent OMA measurements and the EMA measurement to gain a more detailed understanding of the tackled problem. In the first step, classical OMA in standard operational conditions is performed to get an idea of where the problem is located (both in location and frequency). In the second step, OMA during the run-down is conducted, and its results are compared with the results from the first OMA to define precisely where the excessive vibrations are located. Finally, EMA is conducted to get more insight into the structure itself, i.e., when dealing with large machinery impact excitation often fails to excite the vibrations of interest because of low impacting force. This type of behavior suggests non-linearity of the system caused by its complexity, i.e., if the defect is related directly to the impacted structure, it will be appropriately excited. In contrast, if the defect is related to the surrounding elements (e.g., foundation), it will not be excited by the small force of impact excitation. This approach is schematically shown in Figure 1.

### 2.1. Limitations of Modal Analysis

Modal models are a commonly used universal form of describing the dynamics of mechanical structures. They are used to diagnose the condition of the machine under test and can also be used in the health monitoring process. The theoretical assumptions of modal analysis are relatively strict. Following assumptions must be met:

- the linearity of the tested systems—guaranteeing that the response of the system is proportional to the excitation acting on the structure under examination,
- stability of the modal model coefficients during the experiment,
- the Maxwell principle of reciprocity,
- observability of the system,
- ability to measure all the characteristics necessary during the identification, and
- small or proportional attenuation in the tested system.

The above assumptions allow applying the modal superposition principle to the tested system and its presentation as a sum of decoupled harmonic oscillators with one degree of freedom.

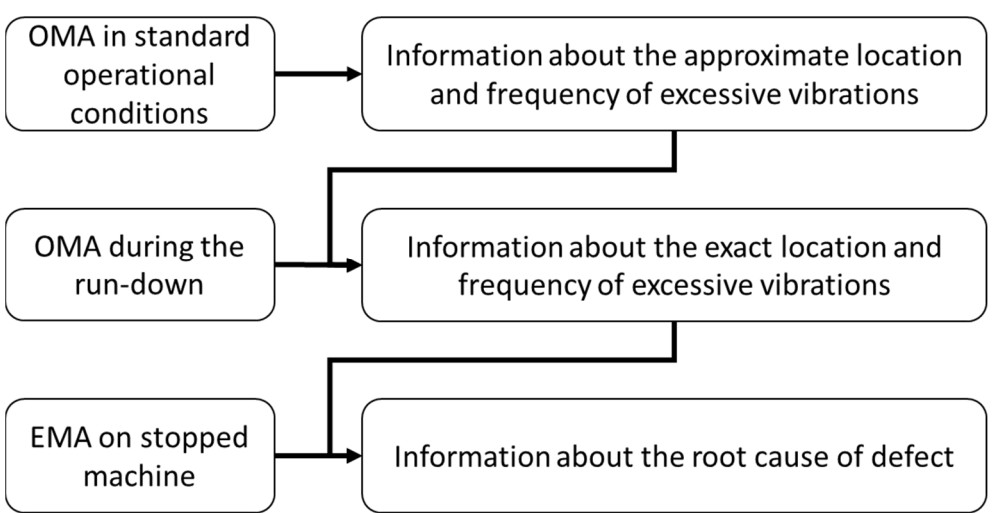

**Figure 1.** Schematic description of the proposed approach.

### 2.2. Comparison of Experimental and Operational Modal Analyses

EMA and OMA are most often used in research practice. In unusual cases for which there are difficulties related to EMA and OMA, the Operational Modal Analysis in the presence of eXogenous inputs (OMAX) model can also be used.

The main differences between EMA and OMA are related to the excitation method of the test object. The EMA assumes the use of an active identification experiment. The tested system is subjected to a controlled and measurable excitation. It is assumed that apart from the controlled (and measured) force, no other input is supplied to the system and all additional effects are treated as disturbances. This approach works very well in laboratory conditions in which the tested object is isolated from its natural working environment and disturbing external factors. The low influence of external disturbances causes that the measurement characteristics are usually of good quality and allow trouble-free identification of modal model parameters. The problem is usually getting the so-called laboratory conditions. Such an operation is often technically complicated, time-consuming, and thus expensive. In the case of objects of large dimensions and great importance, such as generators, their foundations, and other civil engineering structures, it is impossible to isolate the object for the duration of the test. In such a case, any disturbances significantly deteriorate the quality of measurement characteristics while increasing the modal model parameters' estimators' dispersion. Considerable disturbances are usually recorded during tests of power generation machines. There are disturbances related to operations of neighboring machines, usually at the recorded measurement signal level. Measurement characteristics obtained in this way are usually of little use for the estimation of modal model parameters. The situation could be improved by switching off all running machines. In industrial conditions, however, such a situation is infrequent.

Another problem that arises when examining large objects is the provision of an appropriate, exciting force. In the classical EMA, three types of forcing waveforms are most often used: impulse, random noise, and signals composed of sets of properly prepared harmonic functions. The impulse excitation is most often used during preliminary tests. The experiment is carried out in a much shorter time than other methods of forcing, but the measurement characteristics are usually of lower quality. The other two methods of providing a controlled input require additional tools generating vibrations, the so-called shakers. In industrial practice, the use of shakers is significantly limited. Systems of this type that allow the generation of exciting forces at an appropriate level are expensive and

inconvenient to use due to their large size and weight. Another difficulty related to the vibration excitation, occurring primarily in the case of large objects forced by impulse signals, is only the local impact of the excitation on the structure under study. On the one hand, the force which acts on the structure should be large enough to allow the response to be registered at any point on the structure. On the other hand, it cannot be destructive to the object. In the practice of modal research of large power generators and their foundations, it turns out that the response signals are often undetectable due to their masking by disturbances. It is often caused by the inability to isolate the object on which the measurements are made from the environment and the high level of disturbances related to operation of neighboring machines.

Another solution is to test the machines during their regular operation, i.e., using the OMA method. As mentioned earlier, the main difference to the EMA methods is related to object enforcement. OMA assumes that the object is forced only by random white noise with a flat spectrum in the analysis scope. The estimation of modal model parameters is based only on the response of the tested system which curves to the natural operational excitation. Such a situation has several consequences, both positive and negative. First, in most cases, it is not possible to measure the exciting force. Acting forces are usually not tied to a specific place on the structure but have some spatial distribution. In practice, it is not possible to measure operational forces directly. It is sometimes possible to estimate them, e.g., by applying reverse identification methods, but in this case, it is necessary to have a verified model of the system. Another aspect, especially related to operation of energetic machines, is the immense diversity of dynamic conditions during regular operation and transient states. Power machines are designed so that during stable operation they are located at an optimal point on the frequency characteristic. Therefore, in most cases, key moments of energetic machines are the transitional states related to run-up or run-down. These states carry a lot of information about dynamic parameters of the machine, and often, whenever possible, they can be used to identify model parameters.

From the OMA point of view, the modal model should be estimated, based on stationary conditions measurements. Hence, there is often a situation in which, during regular operation, the operational forcing does not allow a sufficiently good forcing of even the essential poles of the system which are excited, e.g., in transient states. Consequently, it can cause difficulties in identification and the inability to detect poles that are weakly forced during regular operation. The modal model obtained as a result of OMA is related to a specific operating point of the tested machine, and therefore its generalization often does not give good results.

Another disadvantage related to the use of OMA in the case of testing energetic machines is the assumption of only a random input acting on the tested system. This assumption causes those harmonic disturbances related to, e.g., operation of other rotating machines located nearby are treated as additional poles of the estimated model. Thus, it is necessary to inspect the parameters of the obtained modal model and recognize and remove redundant poles appearing in the estimation results.

Another possibility in the researcher's hands is integrating the features of OMA and EMA methods in the form of the OMAX model. Such a structure allows for both the controlled forcing of object vibrations and the use of natural operational excitation. It seems that this form of the modal model is best suited for identifying parameters of energetic machines.

The above considerations show that the issue related to obtaining modal models of power generation machines is not trivial and requires a lot of attention and experience, both measurement and practical knowledge during the estimation of modal model parameters.

### 2.3. Basics of Estimating Parameters of Modal Models

Estimation of modal model parameters can be performed in various ways. The most popular methods are based on the time or frequency domain. In the first case, the basis for parameter estimation is the time history of the system's input and response or

directly the waveform of the impulse response function. In the case of estimation methods implemented in the frequency domain, the basis for parameter estimation is the waveforms of the excitation and response spectra or the measured waveforms of frequency transfer functions. In practical applications in diagnostics, the primary methods are classical methods implemented in the frequency domain. It is dictated that it is possible to limit the frequency band to those frequencies in which changes in the vibration waveforms during the operation process are observed.

Due to the size and structure of the model, estimating modal parameters can be divided into methods for systems with one degree of freedom and systems with multi-degrees of freedom. It is assumed that the former behaves like a simple harmonic oscillator in the vicinity of a given natural frequency. The influence of the remaining vibration modes is negligible. These methods can be used for systems with low damping, for which the coupling between particular modes of vibration is negligible. In other cases, methods for multiple degrees of freedom are used.

Time-domain methods are, in principle, multi-degrees of freedom methods. Having analyzed the formulas for the impulse responses of the linear system for the excitation at the $j$th point and the response at the $i$th point it can be seen that:

- poles of the system do not depend on the position of the excitation and the response measurement and can be estimated based on each impulse response (a similar statement can be justified for frequency characteristics),
- mode shapes do not depend on the location of the $j$th point on the structure,
- the input coefficient of participation of a given form in the response does not depend on the position of the $i$th point of response measurement.

Estimation procedures based on the above assumptions result in estimates of global modal parameters determined by methods of estimating global modal parameters.

In the case of analyses performed in the time domain, there can be a distinguished group of methods based on the knowledge of system model in the form of a regression series, most often of the AutoRegressive Moving Average with eXogenous input (ARMAX) type. The analyzer must have some research experience to enable him to choose a method. When choosing, one should pay attention to the following aspects:

- the complexity of numerical calculations,
- range of tested frequencies,
- damping values in the system,
- type of experiment used.

Estimation methods implemented in the time domain, from the point of view of numerical calculations, are better conditioned due to the specific frequency characteristics. From the numerical point of view, the approximation by the least-squares method in the frequency domain is more difficult to implement due to the possibility of local minima. For this reason, time-domain estimation methods are more effective for noisy data. While using them additional errors due to spectral leakage, aliasing or the use of time windows, etc., are avoided. On the other hand, frequency-domain methods offer a more straightforward possibility of data averaging used to remove noise with an average value of zero from the measured waveforms.

## 3. Description of Experiments

The machine under investigation is identified with an excessive vibration level on one of the bearings during standard operational conditions. The source of these vibrations is unknown, so the investigation is divided into three parts, i.e., OMA in standard operational conditions, OMA during the run-down, and EMA when stationary. These analyses are shown in the following subsections.

### 3.1. OMA in Standard Operational Conditions

OMA in standard operational conditions was conducted to identify the frequencies at which highest vibration level is located and how Operational Deflection Shape (ODS) at identify frequency looks like. To achieve this, a dense measurement grid is selected, as presented in Figure 2.

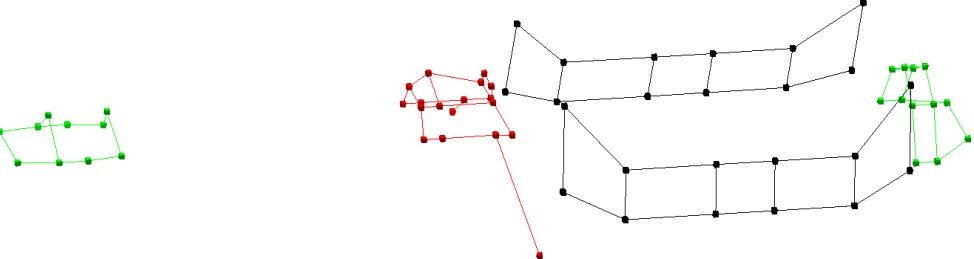

**Figure 2.** Measurement grid for OMA in standard operational conditions. Green and red components are related to bearing pedestals, while black components are related to the generator casing. Points encircled with green eclipse are related to the bearing with excessive vibration levels.

The measurement grid is composed of 64 points measured in three orthogonal directions. The leftmost green component is related to one of the bearings. The center red component is related to two bearings in one bearing pedestal. Points encircled in green eclipse are related to the bearing with excessive vibration levels. Black components are related to the generator casing. The rightmost green components are related to the last bearing.

LMS SCADAS III with 28 channels is used as a data acquisition system, along with 8 3-dimensional (3D) accelerometers PCB 356A16 and 2 1-dimensional (1D) accelerometers PCB 333B30 for capturing acceleration signals. Performing measurements in the entire network of measurement points required 8 partial experiments. During the measurements, the 3D accelerometers were moved during the individual partial experiments, and the 1D accelerometers were used as a reference. The sampling frequency is 1000 Hz, and the acquisition time is 300 s. The averaging window is set to 4 s length to obtain a frequency resolution of 0.25 Hz, and the Hanning window is used for averaging procedure for Cross-Power Densities (CPD) calculation. Data acquisition is made at a constant speed of 3000 RPM, which is the operating speed for the considered machine. The machine is loaded at half of the capacity.

### 3.2. OMA during the Run-Down

OMA during the run-down is done to identify the ODS and corresponding resonant frequencies. Once the ODS are found, a comparison with the ODS from the first experiment will point to the particular resonant frequency, causing the excessive vibration levels. The operational schedule in the plant does not allow to do more than one run-down. Therefore, only a single partial measurement was done, i.e., the measurement grid is reduced to 8 points, as shown in Figure 3.

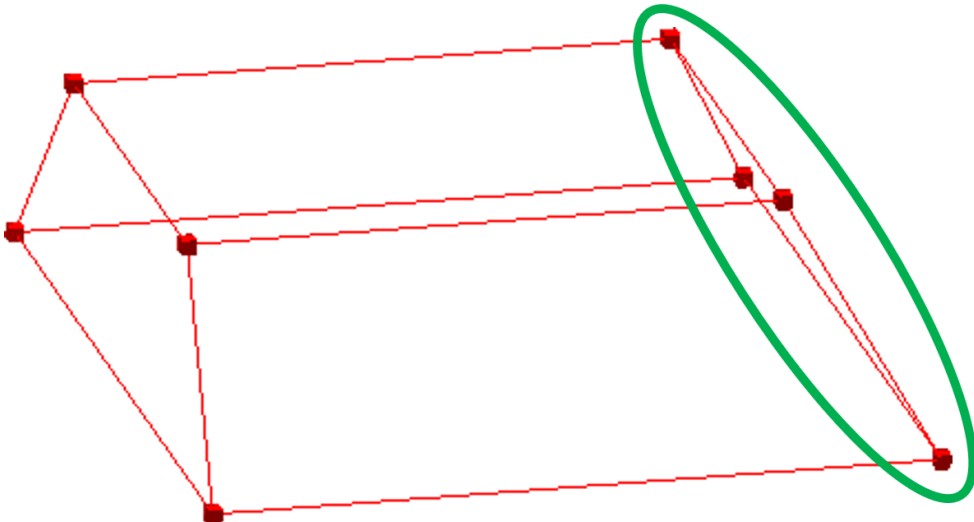

**Figure 3.** Measurement grid for OMA during the run-down. Points encircled with green eclipse are related to the bearing with excessive vibration levels.

Run-down started at 3000 RPM and slowed down to 0 RPM over 40 min. This recording is used to calculate the CPD with the same parameters as in OMA in standard operational conditions.

### 3.3. EMA on Stopped Machine

EMA was done on a stopped machine to identify the Mode Shapes (MS) related to the bearing pedestal. EMA is done with controllable and measurable excitation, and it is done using the modal hammer PCB 086D20. The excitation force is not high enough to excite the entire structure. Thus the excitation is done on the bearing pedestal itself, and the measurement grid is limited to the bearing pedestal as well, as shown in Figure 4.

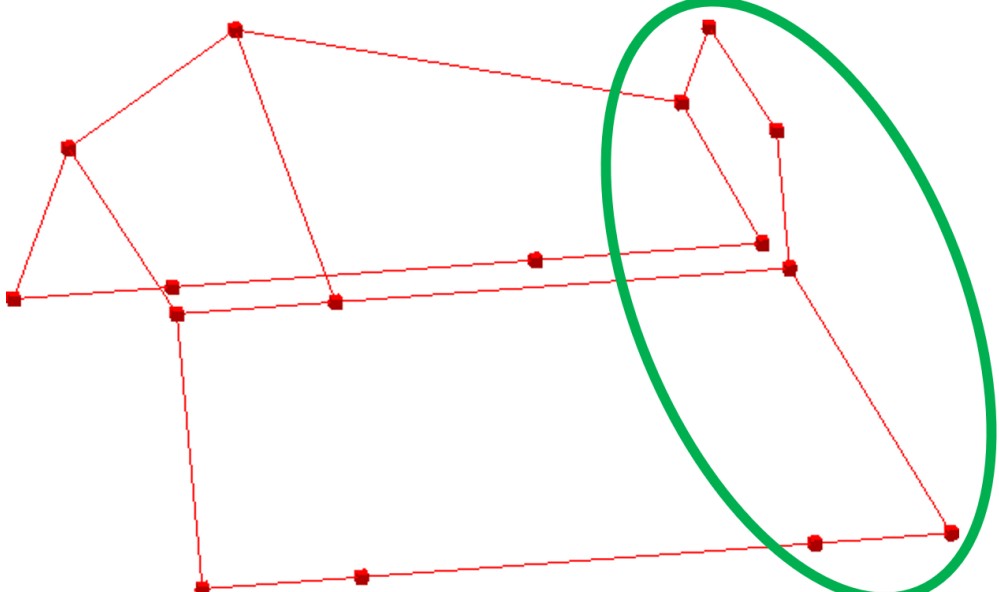

**Figure 4.** Measurement grid for EMA when stationary. Points encircled with green eclipse are related to the bearing with excessive vibration levels.

In this experiment, analysis is limited to 16 measurement points. Thus 2 partial experiments are conducted. For each partial experiment, 20 averages are done to minimize

the external noises' influence on obtained results. The acquisition time window is set to 4 s to obtain the Frequency Response Function (FRF) with 0.25 Hz frequency resolution.

## 4. Results

In this section results from OMA in standard operational conditions, OMA during the run-down and EMA when stationary are presented and analyzed.

### 4.1. OMA in Standard Operational Conditions

Data quality analysis is conducted before actual analysis. This is done by comparison of reference signals from all partial experiments. Calculated Power Spectral Density (PSD) functions are shown in Figures 5 and 6. It is clearly shown that the obtained PSD functions are similar. This confirms the similar operational conditions in all of the partial experiments.

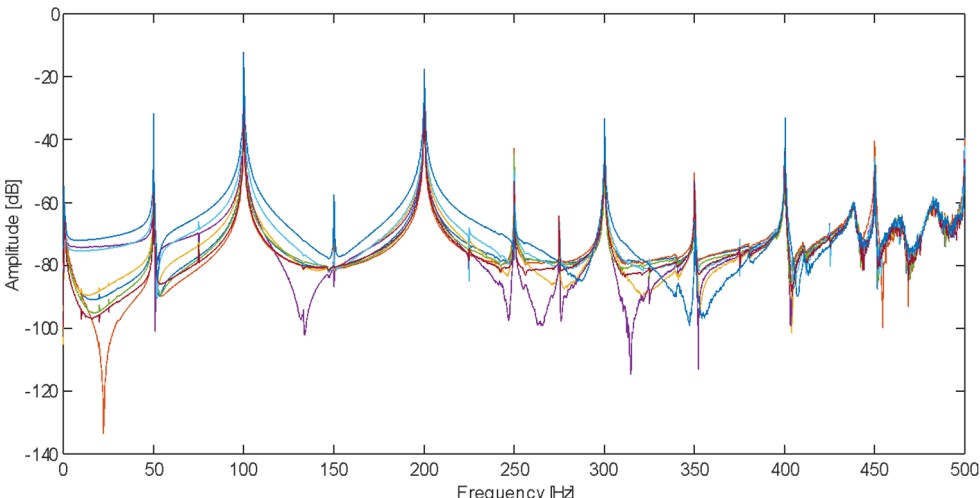

**Figure 5.** PSD functions at the first reference point from all partial experiments.

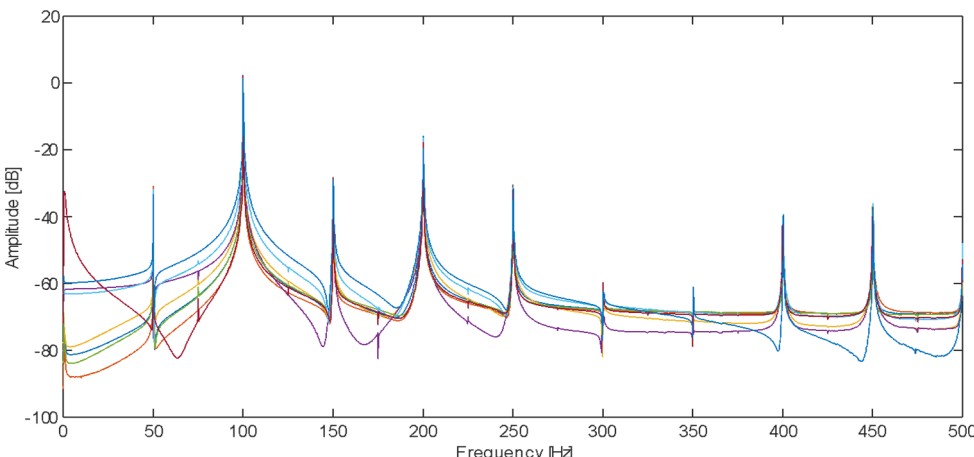

**Figure 6.** PSD functions at the second reference point from all partial experiments.

Based on all acquired CPD functions, the SUM indicator is calculated as shown in Figure 7. This indicator shows that the highest amplitude of vibrations is observed in 100 Hz frequency.

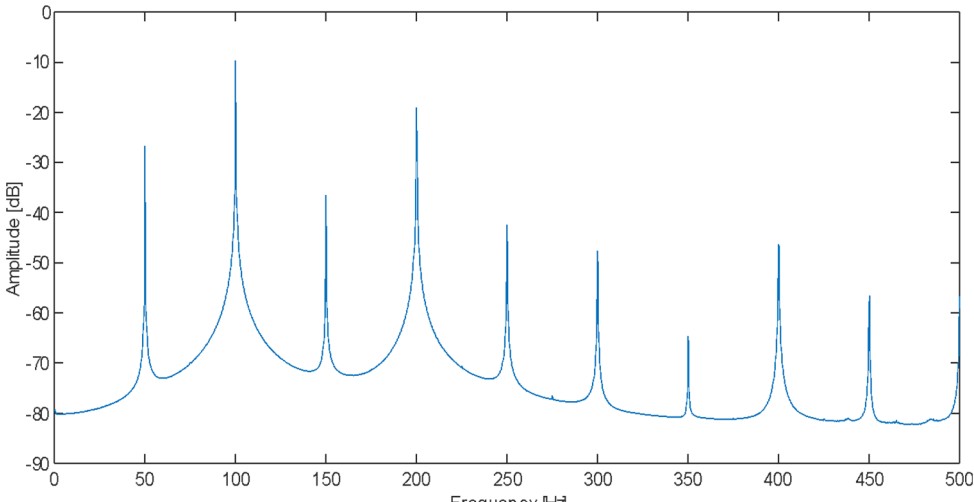

**Figure 7.** SUM indicator from CPD functions acquired during standard operating conditions.

Modal parameter estimation is conducted in LMS Test.Lab Operational Modal Analysis module with use of the OMAX algorithm. A list of identified frequencies and damping ratios is given in Table 1.

**Table 1.** List of identified frequencies and damping ratios from OMS in standard operational conditions.

| Mode No. | Frequency [HZ] | Damping Ratio [%] | Comment |
| --- | --- | --- | --- |
| 1 | 50.01 | 0.00 | Harmonic of rotational speed |
| 2 | 75.01 | 0.01 | Natural frequency |
| 3 | 100.06 | 0.00 | Harmonic of rotational speed |
| 4 | 125.01 | 0.01 | Natural frequency |
| 5 | 133.17 | 0.24 | Natural frequency |
| 6 | 150.09 | 0.00 | Harmonic of rotational speed |
| 7 | 199.96 | 0.01 | Harmonic of rotational speed |
| 8 | 234.27 | 2.56 | Natural frequency |
| 9 | 250.05 | 0.01 | Harmonic of rotational speed |
| 10 | 300.06 | 0.02 | Harmonic of rotational speed |
| 11 | 350.19 | 0.01 | Harmonic of rotational speed |
| 12 | 400.20 | 0.01 | Harmonic of rotational speed |
| 13 | 438.64 | 0.32 | Natural frequency |
| 14 | 450.07 | 0.01 | Harmonic of rotational speed |
| 15 | 464.95 | 0.39 | Natural frequency |

Considering the SUM indicator shown in Figure 7 and the list of identified frequencies given in Table 1, it seems that the highest contribution to vibration levels is in 100.06 Hz frequency. Identified ODS at this frequency is shown in Figure 8 at its two extreme positions.

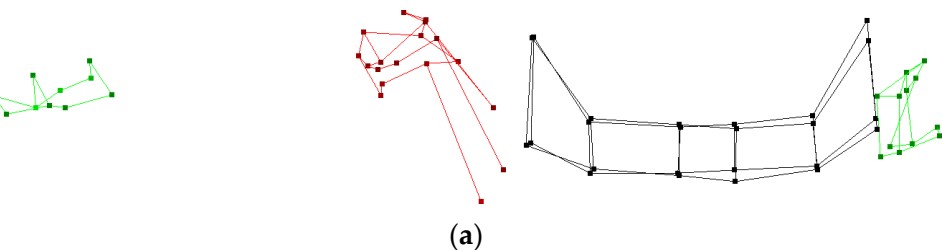

**(a)**

**Figure 8.** *Cont.*

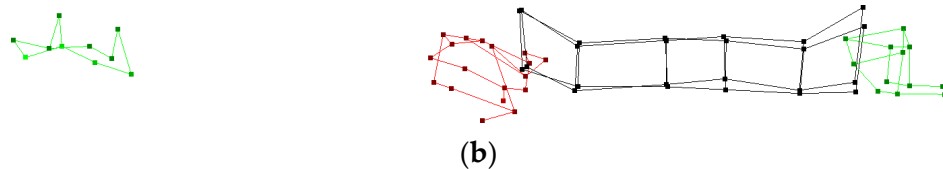

**(b)**

**Figure 8.** Identified ODS at frequency 100.06 Hz: (**a**) first extreme position; (**b**) second extreme position.

It is clearly shown that the right side of the center bearing pedestal has higher deformation amplitudes than its left side. This ODS is responsible for excessive vibration levels on a bearing of interest. It is important to note that ODS could be a composition of motions of several MS.

### 4.2. OMA during the Run-Down

For the case of OMA during the run-down, only 1 partial experiment is done, so there is no need to check for the similarity of consecutive experiments. SUM indicator is shown in Figure 9.

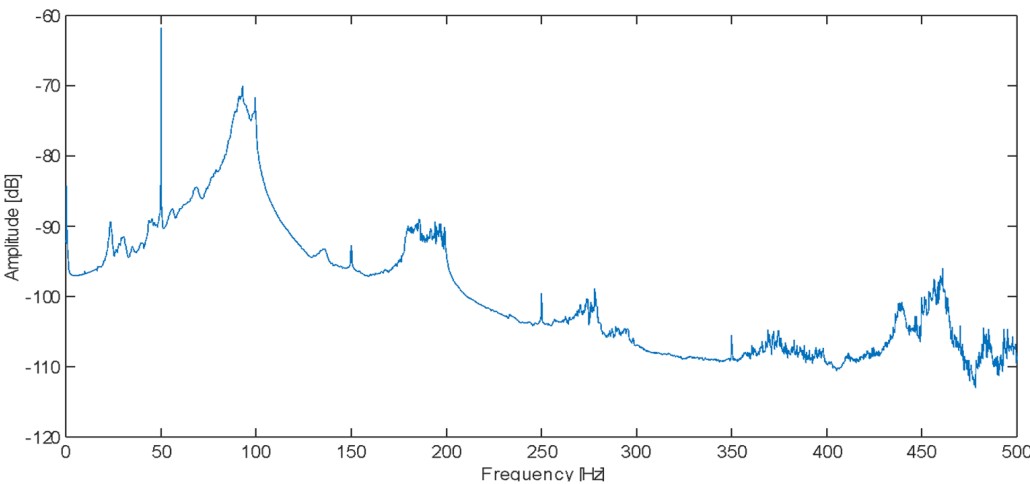

**Figure 9.** SUM indicator from CPD functions acquired during run-down.

Modal parameter estimation is conducted in LMS Test.Lab Operational Modal Analysis module with use of the OMAX algorithm. A list of identified frequencies and damping ratios is given in:

Considering the SUM indicator shown in Figure 9 and the list of identified frequencies given in Table 2, the most interesting ODS frequencies are 90.57, 91.42, 94.80, and 98.91 Hz. Visual comparison of these ODS with the ODS given in Figure 8 pointed to ODS located in frequency 98.91 Hz, shown in Figure 10.

**Table 2.** List of identified frequencies and damping ratios from OMS during the run-down.

| Mode No. | Frequency [HZ] | Damping Ratio [%] | Mode No. | Frequency [HZ] | Damping Ratio [%] | Mode No. | Frequency [HZ] | Damping Ratio [%] |
|---|---|---|---|---|---|---|---|---|
| 1 | 23.36 | 2.74 | 9 | 98.91 | 0.64 | 17 | 286.49 | 0.10 |
| 2 | 33.81 | 3.44 | 10 | 133.46 | 2.03 | 18 | 295.06 | 0.18 |
| 3 | 44.99 | 3.24 | 11 | 170.09 | 0.76 | 19 | 356.79 | 0.22 |
| 4 | 69.03 | 2.21 | 12 | 180.42 | 0.33 | 20 | 373.78 | 0.13 |
| 5 | 79.15 | 4.80 | 13 | 183.54 | 0.57 | 21 | 396.47 | 0.11 |
| 6 | 90.57 | 2.00 | 14 | 198.07 | 0.30 | 22 | 411.61 | 0.10 |
| 7 | 91.42 | 3.09 | 15 | 257.32 | 0.41 | 23 | 440.36 | 0.47 |
| 8 | 94.80 | 1.08 | 16 | 273.25 | 0.11 | 24 | 458.01 | 0.34 |

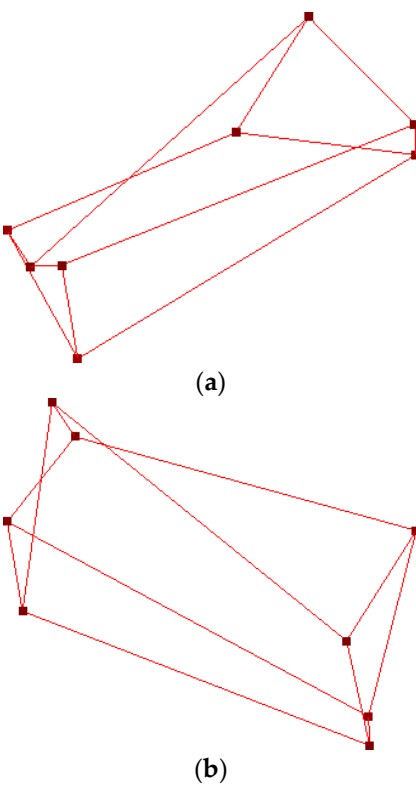

(**a**)

(**b**)

**Figure 10.** Identified MS at frequency 98.91 Hz: (**a**) first extreme position; (**b**) second extreme position.

It is clearly shown that ODS's motion from standard operating condition (Figure 8) and ODS from run-down (Figure 10) are similar. This links the root cause of the excessive vibration levels during the standard operating conditions with the structural dynamics.

### 4.3. EMA on Stopped Machine

It is important to note that these measurements are done during the regular operation of the power plant and similar machines operating nearby. Thus, the environmental noise is exceptionally high. The most important frequencies are related to the vicinity of 100 Hz, as found in previous sections. Therefore, analysis of these frequencies is of primary interest. Firstly, Power Spectral Density (PSD) is checked between 2 partial experiments to check the similarity between excitation levels, as shown in Figure 11.

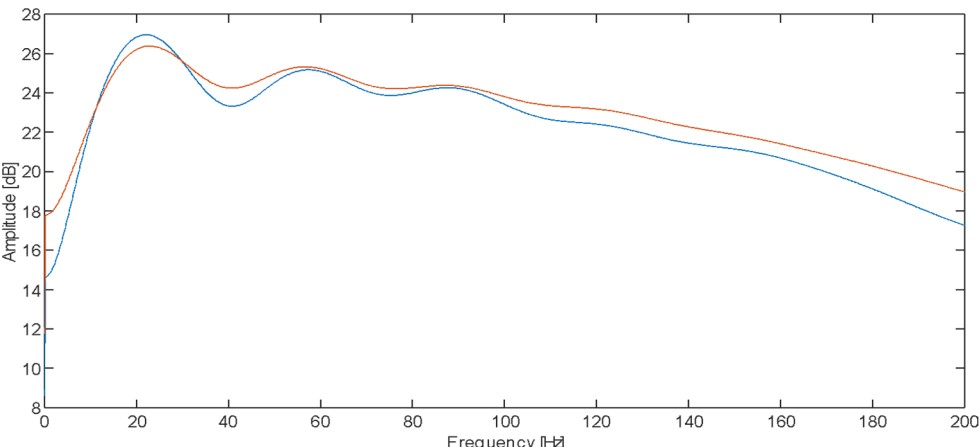

**Figure 11.** PSD of two excitation signals from EMA when stationary experiment. Similarity of these curves present the repeatability of the EMA experiments.

SUM indicator for this analysis is shown in Figure 12. It is important to note that many spikes present on this graph are related to environmental noise, i.e., vibrations from the machine operating nearby.

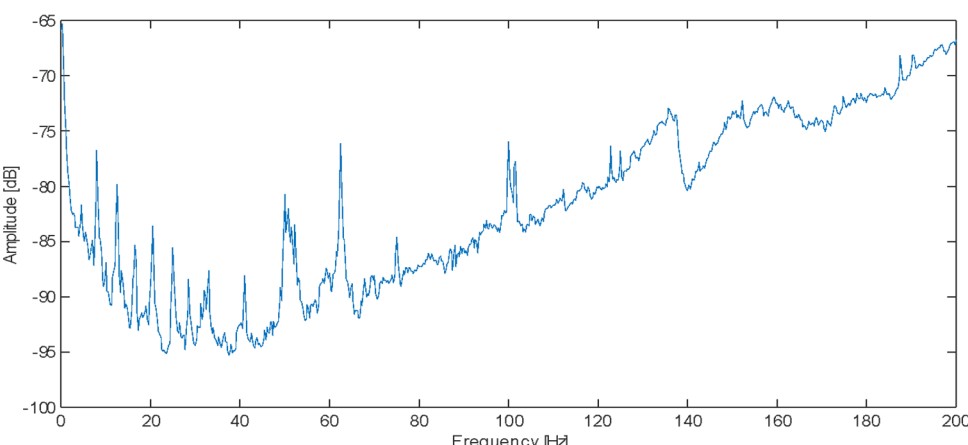

**Figure 12.** SUM indicator from FRF for EMA when stationary.

In a frequency range up to 200 Hz, 95 natural frequencies were identified. Among these frequencies, there are a few close to 100 Hz, but their MS are different from the ODS reported in previous sections. This is most likely related to the modal hammer low excitation force compared to operational excitation. The operational excitation, forced vibration of foundation under the machine, is the cause of the excessive vibration levels.

## 5. Summary of the Results

An engineering problem of bearing pedestal with high amplitude vibrations was tackled using the novel approach combining two OMA measurements and EMA. The ODS at 98.91 Hz was identified and related to the foundation vibrations under the operating machine.

Based on the first conducted analysis, i.e., OMA in standard operating conditions, it was concluded that the excessive vibration level was related to the second harmonic of operating speed and was in the bearing pedestal. It was still unclear what the resonance frequency was—as it was above or below 100 Hz.

The second analysis, i.e., OMA during the run-down, helped to identify four resonant frequencies below 100 Hz. Among these four frequencies, only one ODS resembled the ODS identified in the first analysis. At this stage, it was already evident what resonance frequency was causing the problem. However, it was still uncertain if the problem was related to the machine itself or the foundation underneath.

The third analysis, i.e., EMA on the stopped machine, was unable to identify the problem, which was most likely related to low input excitation compared to operational excitation. If excessive vibration levels were related to the machine itself, one of the identified MS would resemble the ODS identified in the previous analysis. Thanks to elimination, it was concluded, based on EMA, that the problem was located in the foundation of the machine.

## 6. Final Conclusions

The paper presents a new hybrid approach to the assessment of the dynamic condition of large rotating machines used in the power generation industry. The combination of many variants of different MA types (EMA, OMA based on stationary responses, OMA based on transient responses) allowed to identify a potential source of the problem of increased vibration level. In the literature, one can find the use of combining EMA and OMA for the modal identification of objects. This combination is most often used to find the MS scaling factors obtained on the basis of OMA [38,39]. In the presented work, EMA was used for geometric verification of MS obtained by OMA analyzes.

Such an innovative application gives additional information to the researcher and allows for a more confident interpretation of the results. The proposed approach should be adopted by users of power generation assets and other critical machinery, alike. Better modal models can improve knowledge of the dynamic properties of machine. In case of a deteriorated dynamic behavior it can shorten and improve looking for a root cause of a problem. Eventually, it can help to increase availability of critical assets.

**Author Contributions:** Conceptualization, K.D., K.M., P.K. and T.B.; methodology, K.D. and K.M.; software, K.D.; validation, K.D., K.M., P.K. and T.B.; formal analysis, K.D. and K.M.; investigation, K.D. and P.K.; resources, T.B.; data curation, P.K.; writing—original draft preparation, K.D.; writing—review and editing, K.M., P.K. and T.B; visualization, K.D.; supervision, K.M.; project administration, T.B.; funding acquisition, K.M. and T.B. All authors have read and agreed to the published version of the manuscript.

**Funding:** The APC was funded by AGH, Robotics and Mechatronics Department.

**Institutional Review Board Statement:** Not applicable.

**Informed Consent Statement:** Not applicable.

**Data Availability Statement:** Not applicable.

**Conflicts of Interest:** The authors declare no conflict of interest.

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
