# Peer review of "Multi-Variant Modal Analysis Approach for Large Industrial Machine"

_energies, doi:10.3390/en15051871_

Round 1
Reviewer 1 Report
The authors presented an approach of combining the results of two Operational Modal Analyses and Experimental Modal Analysis to improve the obtained models' quality. The subject is interesting, but the paper needs to be improved. Also, the language needs to be refined. Other comments are:
- The abstract must be revised by a proofread, and it is needed to be written in a professional way;
- The introduction needs the presentation of more background, and more recent references. Also, the summarization of the results is needed;
- The numbering of the sections must be fixed;
- The second section needs to be clearly written, and more subsections are needed. The methodology is not clearly described now;
- The description of the results needs to be renamed to "results".
- Also, the results needs to be discussed with the literature.
As the paper is written, it does not have a clear value.
Author Response
The authors presented an approach of combining the results of two Operational Modal Analyses and Experimental Modal Analysis to improve the obtained models' quality. The subject is interesting, but the paper needs to be improved. Also, the language needs to be refined. Other comments are:
- The abstract must be revised by a proofread, and it is needed to be written in a professional way;
Authors have used professional proofread service
- The introduction needs the presentation of more background, and more recent references. Also, the summarization of the results is needed;
In total 9 new references were added and introduction was extended. Summarization of the results section was extended.
- The numbering of the sections must be fixed;
Corrected.
- The second section needs to be clearly written, and more subsections are needed. The methodology is not clearly described now;
After correction of the introduction part, Authors think that this section will now make more sense. Additionally more sections were added to divide the blocks of text
- The description of the results needs to be renamed to "results".
Corrected.
- Also, the results needs to be discussed with the literature.
This was added in last section
As the paper is written, it does not have a clear value.
Reviewer 2 Report
The authors have made an important contribution through this work. Th combination of different modal analysis to ascertain the cause of failure is novel.
A few minors comments are suggested to be followed for improved impact of the manuscript:
- The text following the figure and the caption of the figure is almost getting merged in each case. Please provide adequate gap. This comment pertains to all figures in the manuscript.
- Abbreviations of EMA and OMA for Experimental and Operational is not required. Why use three letter abbreviation for one word? Even if earlier it is there in any literature, there is no justification to continue this practice.
- Can EMA be done for Standard OPerational conditions and Run-down and OMA for stopped machine (in addition to 2.1, 2.2, and 2.3) ?
Author Response
The authors have made an important contribution through this work. Th combination of different modal analysis to ascertain the cause of failure is novel.
A few minors comments are suggested to be followed for improved impact of the manuscript:
- The text following the figure and the caption of the figure is almost getting merged in each case. Please provide adequate gap. This comment pertains to all figures in the manuscript.
Corrected.
- Abbreviations of EMA and OMA for Experimental and Operational is not required. Why use three letter abbreviation for one word? Even if earlier it is there in any literature, there is no justification to continue this practice.
EMA refers to Experimental Modal Analysis and OMA refers to Operational Modal Analysis, these are three words abbreviations and they are very different from each other (EMA vs OMA), this will be discussed in more detail in response to following comment
- Can EMA be done for Standard OPerational conditions and Run-down and OMA for stopped machine (in addition to 2.1, 2.2, and 2.3) ?
EMA as noted in above response is Experimental Modal Analysis, while OMA is Operational Modal Analysis. The difference between the two is in the type of excitation force provided. In the case of the EMA, the excitation force is provided by personnel executing the measurement, i.e., force is applied to structure by people, while in case of OMA, we are dealing with the force being applied by Operational forces, so by the machine itself. This is a grate difference between the two, as in case of EMA we are able to measure these forces and use them in calculation of Frequency Response Functions (FRF), whereas in case of OMA it is impossible to measure these forces, so we are only able to measure Power Spectral Densities (PSD) and Cross Power Densities (CPD). It is much easier to conduct Modal Analysis with FRFs, but there are limitations to that, as provided excitation might be too small to properly excite the structure, whereas in case of Operational excitation, there is enough force to excite entire structure, but we have to work with PSDs and CPDs, this is the reason why this research is very problematic.
Reviewer 3 Report
The manuscript titled "Multi-Variant Modal Analysis Approach for Large Industrial Machine" has been reviewed carefully. After careful review of this work, I recommend that the authors revise this work and submit it for re-review. In my opinion, a careful revision of the English language should be carried out as there currently are some unclear sentences. These grammatical and syntax errors make the paper difficult and impossible to understand in some sections. Also, the most critical concern is the lack of verification and validation of the model, which should be considered carefully.
- The abstract needs to be revised. It is better to discuss the subject's generalities first and then explain the materials and method of the research.
- A careful revision of the English language should be carried out. There are currently several unclear sentences grammatical and syntax errors that make the paper difficult and even impossible to understand in some sections.
- In the introduction, the novelty and contribution of this work are not apparent. The introduction of the paper is poor and generally not organized. That is, there is not a good flow between different paragraphs and sections in the introduction. Also, unnecessary sections in the introduction (as sections 1.1 to 1.3) can be summarized. The following references are recommended:
- Pile-soil interaction determined by laterally loaded fixed head pile group." Geomechanics and Engineering 26.1 (2021): 13-25.
- Azizi, F., Vadiati, M., Asghari Moghaddam, A., Nazemi, A., & Adamowski, J. (2019). A hydrogeological-based multi-criteria method for assessing the vulnerability of coastal aquifers to saltwater intrusion. Environmental Earth Sciences, 78(17), 1-22.
- Predicting the uniaxial compressive strength of oil palm shell lightweight aggregate concrete using artificial intelligence‐based algorithms. (2022). Structural Concrete.
- Rezaei, K., Pradhan, B., Vadiati, M., & Nadiri, A. A. (2021). Suspended sediment load prediction using artificial intelligence techniques: comparison between four state-of-the-art artificial neural network techniques. Arabian Journal of Geosciences, 14(3), 1-13.
- The manuscript lacks a good flow between different sections. Also, it does not highlight its contribution to the field.
- The authors are encouraged to present some performance evaluator indices to compare the results of numerical and experimental analysis.
- The legend of all figures should be provided (for example, Fig. 11).
- It is recommended to present a models programming table in the materials and methods section.
- The limitations of the present study should be added to the paper.
- Executive suggestions from the present study are essential for the application of engineers and designers.
- It seems that conclusions are observations only, and the manuscript needs thorough checking for explanations given for results. The authors should interpret more precisely around results argument.
Author Response
The manuscript titled "Multi-Variant Modal Analysis Approach for Large Industrial Machine" has been reviewed carefully. After careful review of this work, I recommend that the authors revise this work and submit it for re-review. In my opinion, a careful revision of the English language should be carried out as there currently are some unclear sentences. These grammatical and syntax errors make the paper difficult and impossible to understand in some sections. Also, the most critical concern is the lack of verification and validation of the model, which should be considered carefully.
- The abstract needs to be revised. It is better to discuss the subject's generalities first and then explain the materials and method of the research.
Corrected
- A careful revision of the English language should be carried out. There are currently several unclear sentences grammatical and syntax errors that make the paper difficult and even impossible to understand in some sections.
Authors have used professional proofread service
- In the introduction, the novelty and contribution of this work are not apparent. The introduction of the paper is poor and generally not organized. That is, there is not a good flow between different paragraphs and sections in the introduction. Also, unnecessary sections in the introduction (as sections 1.1 to 1.3) can be summarized. The following references are recommended:
- Pile-soil interaction determined by laterally loaded fixed head pile group." Geomechanics and Engineering 26.1 (2021): 13-25.
- Azizi, F., Vadiati, M., Asghari Moghaddam, A., Nazemi, A., & Adamowski, J. (2019). A hydrogeological-based multi-criteria method for assessing the vulnerability of coastal aquifers to saltwater intrusion. Environmental Earth Sciences, 78(17), 1-22.
- Predicting the uniaxial compressive strength of oil palm shell lightweight aggregate concrete using artificial intelligence‐based algorithms. (2022). Structural Concrete.
- Rezaei, K., Pradhan, B., Vadiati, M., & Nadiri, A. A. (2021). Suspended sediment load prediction using artificial intelligence techniques: comparison between four state-of-the-art artificial neural network techniques. Arabian Journal of Geosciences, 14(3), 1-13.
Suggested references are not quite related to research concerning the Modal Analysis, rather they are related to soft-computing methods and some geomechanics, oil palm sciences, so it seems that this is related to environmental sciences, while our article is related more to the mechanical engineering. In total 9 new references were added, but none of the suggested above as Authors do not think that suggested references are relevant.
- The manuscript lacks a good flow between different sections. Also, it does not highlight its contribution to the field.
General flow of the article was updated in all sections
- The authors are encouraged to present some performance evaluator indices to compare the results of numerical and experimental analysis.
The submitted paper contain only experimental data that is analysed.
- The legend of all figures should be provided (for example, Fig. 11).
Corrected.
- It is recommended to present a models programming table in the materials and methods section.
This study does not contain any numerical study or programmable models.
- The limitations of the present study should be added to the paper.
The limitations of the Modal Analysis are listed in the 2 section, they are as follows:
The theoretical assumptions of modal analysis are relatively strict. Following assumptions must be met:
- the linearity of the tested systems - guaranteeing that the response of the system is proportional to the excitation acting on the structure under examination,
- stability of the modal model coefficients during the experiment,
- the Maxwell principle of reciprocity,
- observability of the system,
- ability to measure all the characteristics necessary during the identification, and
- small or proportional attenuation in the tested system.
- Executive suggestions from the present study are essential for the application of engineers and designers.
This was added in last section
- It seems that conclusions are observations only, and the manuscript needs thorough checking for explanations given for results. The authors should interpret more precisely around results argument.
This was added in last section
Round 2
Reviewer 1 Report
The paper is improved and it can be accepted.
Reviewer 3 Report
The authors addressed my comments.